# Combining Ergonomic Risk Assessment (RULA) with Inertial Motion Capture Technology in Dentistry—Using the Benefits from Two Worlds

**DOI:** 10.3390/s21124077

**Published:** 2021-06-13

**Authors:** Christian Maurer-Grubinger, Fabian Holzgreve, Laura Fraeulin, Werner Betz, Christina Erbe, Doerthe Brueggmann, Eileen M. Wanke, Albert Nienhaus, David A. Groneberg, Daniela Ohlendorf

**Affiliations:** 1Institute of Occupational Medicine, Social Medicine and Environmental Medicine, Goethe-University, 60323 Frankfurt am Main, Germany; christian.maurer.cm@gmail.com (C.M.-G.); maltry@med.uni-frankfurt.de (L.F.); brueggmann@med.uni-frankfurt.de (D.B.); wanke@med.uni-frankfurt.de (E.M.W.); groneberg@med.uni-frankfurt.de (D.A.G.); ohlendorf@med.uni-frankfurt.de (D.O.); 2Institute of Dentistry, Department of Dental Radiology, Goethe-University, 60323 Frankfurt am Main, Germany; w.betz@em.uni-frankfurt.de; 3Department of Orthodontics, Medical Center of the Johannes Gutenberg-University Mainz, 55128 Mainz, Germany; erbe@uni-mainz.de; 4Principles of Prevention and Rehabilitation Department (GPR), Institute for Statutory Accident Insurance and Prevention in the Health and Welfare Services (BGW), 20246 Hamburg, Germany; albert.nienhaus@bgw-online.de

**Keywords:** inertial motion units, wearable sensors, human factors, work place evaluation, ergonomics, kinematic analysis, dentist, dental assistant, dental treatment concept

## Abstract

Traditional ergonomic risk assessment tools such as the Rapid Upper Limb Assessment (RULA) are often not sensitive enough to evaluate well-optimized work routines. An implementation of kinematic data captured by inertial sensors is applied to compare two work routines in dentistry. The surgical dental treatment was performed in two different conditions, which were recorded by means of inertial sensors (Xsens MVN Link). For this purpose, 15 (12 males/3 females) oral and maxillofacial surgeons took part in the study. Data were post processed with costume written MATLAB^®^ routines, including a full implementation of RULA (slightly adjusted to dentistry). For an in-depth comparison, five newly introduced levels of complexity of the RULA analysis were applied, i.e., from lowest complexity to highest: (1) RULA score, (2) relative RULA score distribution, (3) RULA steps score, (4) relative RULA steps score occurrence, and (5) relative angle distribution. With increasing complexity, the number of variables times (the number of resolvable units per variable) increased. In our example, only significant differences between the treatment concepts were observed at levels that are more complex: the relative RULA step score occurrence and the relative angle distribution (level 4 + 5). With the presented approach, an objective and detailed ergonomic analysis is possible. The data-driven approach adds significant additional context to the RULA score evaluation. The presented method captures data, evaluates the full task cycle, and allows different levels of analysis. These points are a clear benefit to a standard, manual assessment of one main body position during a working task.

## 1. Background

For the assessment of the ergonomic risk potential, various methods are available to identify imbalances between workplace conditions and the physiological capabilities of the employee, which are referred to as Ergonomic Risk Assessment Tools (ERATs) [1,2,3,4,5]. The choice of the particular ERAT depends on the risk dimension studied. Common risk dimensions result from applied forces, duration of physical work, or postural variability [6]. The aim of the application of ERATs is to conduct structured and objectified ergonomic analyses of workplaces in order to develop preventive solution strategies based on these findings [7]. For methodical implementation, *observational methods* or *direct measurements* [2,7,8] are frequently used among others. The Rapid Upper Limb Assessment (RULA) [8] is an internationally used popular *observational method*, which is scientifically well evaluated and reliable. RULA investigates the kinematics of body regions such as neck, shoulders, trunk, arms, and hands with a specific focus on force, frequency, and duration. In this process, the ergonomic risk of an occupational activity can be quantified by means of resulting scores. These scores enable simple assessments of hazard potentials including a result visualization via scoring nomenclature. The scores are usually recorded by trained personnel (observer), who document the observed work processes on a paper-pencil basis [9]. A disadvantage, however, is that the observer can hardly assess the entire occupational demands during a complete work process or shift or day at work. This results in a selective investigation of those postures that are performed most frequently, commonly account for more than 10–15% of a task and are the most harmful [10].

In *direct measurements*, the entire work process is quantified by means of various measurement systems, independent of subjective expert assessments. A continuous recording of postures in the daily work routine is made possible by inertial motion units (IMU) [2]. In this study, a kinematic model is created using acceleration sensors and gyroscopes, through which the positions of body segments and joint angles are validly and reliably reproduced on the computer [4,11,12]. Direct measurements are commonly used to examine workplaces in the automotive industry [13], in offices [4], at supermarket checkouts [11], and workplaces with manual activities [12].

In order to utilize the objective and high-frequency kinematic data for ergonomic risk assessments, the information can be applied to scoring models such as the RULA [8,14,15]. This creates a combination of an *observational method* with *direct measurements*—specifically motion detection with inertial sensor technology and the RULA procedure for the ergonomic assessment of workplaces, which was initially described by Vignais et al. in 2017 [7]. Recently, improved approaches combining observational method with direct measurements were published [7,16,17,18]. Here, known weaknesses of the RULA procedure, e.g., missing thresholds for arm abduction, radial or ulnar abduction, as well as head abduction and trunk rotation, were overcome by the ability of the system to quantify angles and subsequently set thresholds. In addition, an innovative feature is the calculation of the scores over time (depending on the recording frequency of the measuring system, in this case 64 Hz). This includes the RULA total score and all sub scores for the right and left half of the body [16]. The total ergonomic load can thus be determined more decidedly and objectively with the result of a more precise and comprehensive ergonomic assessment of workplaces. This combination could also be used to explore possibilities to improve the sensitivity of the newly developed automation of RULA. Because IMU data enable to capture posture with a very high time and space resolution in multiple joints and degrees of freedom, the accuracy of the resulting ergonomic risk score is solely dependent on the generalization level of the of the applied ERATs.

This present analyzing procedure was designed aiming to develop a script that allows an ergonomic risk quantification based on RULA using kinematic data of joint angles and positions of body segments collected by IMU. On this basis, the ergonomic risk can then be assessed using IMU data. To achieve the aim of the study, we implemented the IMU data in the RULA steps. In some steps, modifications to the RULA scores were necessary, since, e.g., no angular ranges are determined in the original version. For simplification, the 15 RULA steps were summarized in five stages. This automatized RULA version offers several possibilities for more or less complex and accurate analysis, which was explored further as “levels of complexity”. This can be useful, when two very similar working conditions are compared and the original RULA scores are too general to detect differences. To present this approach, we used data from the SOPEZ project [14], in which the association of musculoskeletal disorders and ergonomics of dentists and dental assistants was investigated [16,19,20,21,22,23,24]. The intended RULA adjustment will compute a total RULA score for each frame, so that corresponding statistical calculations can be performed on this basis. This manuscript focuses on the methodological development of the approach and not on the evaluation of the sample data.

## 2. Methods

### 2.1. Subjects

For the biomechanical measurements, 15 (12 males/3 females) oral and maxillofacial surgeons (age: 36.2 ± 9.2 years) were invited to participate in this pilot study. The dentists worked as oral and maxillofacial surgeons in established dental practices. The participants had, on average, a working experience of 9.5 ± 8.7 years. Inclusion criteria were right-handedness of the surgeons and the ability to treat in both dental treatment concepts named “treatment concepts 1 and 2” (Figure 1). Exclusion criteria included current injuries to the musculoskeletal system (e.g., herniated discs and spinal injuries), rheumatic diseases, severely restrictive malformations (scoliosis) of the spine or stiffened spinal joints (pathological or surgical), genetically determined muscular diseases, and surgery less than 2 years ago. The study was approved by the Ethics committee of the Department of Medicine of the Goethe University Frankfurt am Main (Germany) (No. 356/17).

### 2.2. Recruitment

The participants were recruited using a flyer, which was distributed in local surgical dental practices. In addition, maxillofacial surgeons working in the Institute of Dentistry of the Goethe University Frankfurt were invited to participate in this study.

### 2.3. Measurement Protocol

To record the posture of the study participant in the different test situations, we used the inertial motion capture system MVN Link from Xsens (Enschede, The Netherlands). For this purpose, the test persons had to wear a measuring suit. In order to measure whole body kinematics, all 17 sensors were attached, as specified by the manufacturer. This IMU system was used to measure the 3D angle of the segments and to calculate the 3D joint angles required to calculate the RULA scores. After this suit has been put on, the recordings took place in each of the two spatial treatment concepts (Figure 1). On a dummy head, which was attached to each of the two treatment concepts, dental treatment activities were carried out in a standardized sequence. In order to have an extern reference, the entire measurement sequence was filmed simultaneously from a bird’s eye view (iPad Air, Apple Inc., Cupertino, CA, USA; resolution: 1080p HD, 120 fps). The 2D camera was, as mentioned, only used as a reference device, to capture the full scenery. Therefore, the measurement system and the camera were synchronized by the MVN Analyze software (Xsens (Enschede, The Netherlands). A standardized surgical task was performed in both dental treatment concepts (Figure 1). The dentists conducted a palatinal and marginal incision in region 16 to 11 in the first quadrant for approximately 60 s.

### 2.4. Measurement System

In this study, kinematic data were collected using the inertial motion capture system MVN Link from Xsens (Enschede, The Netherlands). Therefore, 17 inertial sensors, a transmission pack (body pack), and a battery were attached to a special suit. Each sensor includes a 3D linear accelerometer, gyroscope, barometer, and magnetometer, internally sampling at 1000 Hz [25]. According to the manufacturer, the overall output sampling rate is 240 Hz and the measurement error is specified as ±1%. Compared to optical motion tracking (gold standard), this inertial motion capture system delivers good to excellent data referring to concurrent findings, especially in the frontal and sagittal planes [26,27].

The Xsens system interpolates among others a total of 22 joints with 3 dimensions and data of position and orientation of 23 segments. The manufacturer recommended to record in the “no-level” function, when no floor interactions or positional changes in space were performed. This means that the hip segment is the reference of the individual coordinate system. This has no negative effect on the parameters listed above, which are outputted by the system. Furthermore, as recommended by the manufacturer, all data were subjected to the “HD reprocess” in order to achieve the best possible data quality.

### 2.5. Dental Treatment Concepts

While performing dental treatments according to DIN EN ISO 4073:2009-10, dentists have to remain in specific positions, next to the patient or behind the patient, which allow only minimal posture shifts for 7–12 h. Different dental treatment concepts are used in dentistry worldwide. In this study, the two most commonly used spatial treatment concepts in Germany will be compared. In treatment concept 1, the functional areas of dentist and assistant are divided (“split unit”). Seen from the bird’s eye perspective, the dentist is located in a predominantly seated working position at 9 o’clock, right from the patient. Here, in this main working position, the dentist performs all main working tasks at a deeply reclined patient. The dental assistant, seated at 3 o’clock left from the patient, is responsible for suction and holding tasks (also in a predominantly seated working position), (Figure 1 left). In treatment concept 2, the dentist reaches (while located at a 9 o’clock position in a predominantly seated working position) with the left hand for the instruments located on the treatment center that is arranged at the 12 o’clock position. Then right-handed dentists must transfer the hand pieces to the right hand. The dental assistant (seated at 3 o’clock) is responsible for transferring the instruments, whereby the suction and holding technique corresponds to treatment concept 1 (Figure 1, right).

### 2.6. Rapid Upper Limb Assessment

RULA [1,8] is used to assess the ergonomic risk of workflow processes. The system’s focus is on the body regions such as neck, shoulders, trunk, arms, and hands with a predominantly kinematic approach. Using a series of illustrations of different body postures, an overall “global” posture can be quantified (Figure 2) [8,28,29,30]. Here, the evaluation protocol is divided into three main steps: Step A contains the measurements of the upper arm, lower arm, and wrist. Step B contains the measurements of the neck, trunk, and legs. The posture scores of steps A and B are extended in subsequent measurements by values for static muscle work and force resulting in the “Wrist & Arm Score” and “Neck, Trunk, Leg Score”, respectively. In step C, the total score is calculated, which evaluates the ergonomic risk according to the following classifications [8]:1–2: Posture is acceptable if not maintained.3–4: Further investigation needed. May need changes.5–6: Further investigation and changes needed soon.7:  Investigation and changes required immediately.

### 2.7. Implementation of RULA to IMU-Based Evaluation

All 15 RULA steps are fully implemented in the presented scripts. For illustration purpose only, these 15 RULA steps can be summarized by the five stages presented in Figure 3. The sketch is a simplification of the process highlighting the main processes within each stage. The joint angles are calculated based on the measured angles of the IMUs. Starting from the raw joint angles, for every time point, the specific angle to score mapping is applied in accordance to the RULA definitions/steps (Figure 2) and the required adaptations (Table 1 and Appendix A Table A1).

As shown in Figure 3–stage 1, first, the angle ranges are mapped to the specific score values. For instance, for the shoulder flexion–extension angles between −20 and 20 are mapped to 1, angles between 20 and 45 to 2, angles between 45 and 90 to 3, and values above 90 to 4 (not shown) (Figure 3 stage 1).

At each time point, the values are added up resulting in the score for step 1 for the right shoulder (Figure 3 stage 2). The final kinematic score for the arm and wrist (step 5) is determined based on a look up table combining the scores for step 1 to 4 (Figure 3 stage 3).

By repeating the same procedure for the muscle use and force applied on the neck, trunk, and leg, the scores for the arms (step 8) and neck, trunk, and leg (step 15) are computed for every time point. The final RULA score distribution is obtained based upon the lookup table C (Figure 3 stage 4) [8]. The final RULA score is determined as the median score across all time points represented as overall assessment of the movement (Figure 3 stage 5).

### 2.8. RULA Modifications

The observational scoring models can only partly be applied to the objective and continuous data from the IMU sensors. Several parts of the RULA protocol that do not consider quantitative limits need to be adapted (Table 1).

### 2.9. Analysis of Levels of Complexity

So far, the implementation of the IMU data into the RULA worksheet has been presented. Further, the newly developed data offer different possibilities for more or less complex and accurate analysis, which are explored in the present section.

One can identify five different levels of complexity based on the variables. The five levels ordered by the least complex to the most complex one are:RULA scoreRelative RULA score distributionRULA steps scoreRelative RULA steps score distributionRelative angle distributions.

This order is obviously reversed to the calculation procedure, as the calculation of the RULA score starts from the highest resolution (IMU raw data) and ends in one score per treatment concept per subject. However, the most general answer is gathered from the RULA score. If this gives a large difference between two conditions, the following will give a difference as well.

The first level **RULA score** is one number per subject per condition. The ordinal distributed numbers can take the values 1 to 7.

The second level, the **relative RULA score distribution**, is calculated as the relative occurrences of each value of the RULA score for the full measurement time. Therefore, every value can be seen as a variable, with possible values from 0% to 100%. With an assumed resolution of 1% this is a total number of 700 possible values.

The third level, the **RULA steps score**, is the median across the time-dependent steps. For the ergonomic risk assessment of dentists, only the 1st step–shoulder, 2nd step–elbow, the sum of the 3rd step and 4th step–wrist, the 9th step–neck, and the 10th step–trunk are considered important, as they involve the kinematics of the arms, head, and trunk. The first three are calculated for both sides. This results in eight variables.

The fourth level of complexity is the **relative RULA steps score occurrence**. In analogy to the RULA score, every score can be seen as a variable, and the resolution of the relative occurrence is set to 1%. This leads to 56 variables with 5600 possible values.

In Table 1, 1st step is calculated based on three angles. The 2nd step is based on two angles, the sum of the 3rd and 4th step is based on two angles, the 9th step is based on three angles, and the 10th step is also based on three angles. As the first three parts are calculated for both sides, one ends up with 20 variables. Assuming an angle resolution of 1° and an average range of motion across all the joints of 70 degrees, one ends up with 1400 possible angle values in total.

### 2.10. Statistical Data Analysis

Postprocessing of the IMU data has been conducted using of the Xsens software. After exporting the joint angles all-further processing steps were conducted in custom written files by the MATLAB^®^ vR2018a software (The MathWorks Inc., Natick, MA, USA) (Appendix A). To estimate the separation, a maximum resolution of 1% was considered for the relative score occurrence or relative angle occurrence and a maximum of 1° was considered for the angle distribution. The complexity was estimated based on the number of variables. The numbers were calculated as the product of independent scores times score levels or as the number of angles times the number of possible angles with a difference of 1°.

The non-parametric Wilcoxon rank sum test was used to test for significance in treatment concept 1 versus treatment concept 2. For sets with multiple comparison, a Bonferroni correction was applied. The relative angle distribution was compared using statistical parametric mapping (SPM) [31]. Again, a Bonferroni correction was applied to contribute for the multiple angles we looked at. The alpha value for significance was set to 0.05.

## 3. Results

The complexity versus resolution is plotted in Figure 4. The first level of complexity (RULA score) consisted of one variable with seven different numbers. Therefore, it had the lowest complexity and the lowest resolution. On the other side of the spectrum, the fifth level of complexity (relative angle distribution) had the highest complexity and the highest resolution.

For every set of the five levels of complexity, statistical analysis was performed in order to compare treatment concept 1 versus concept 2 (Table 2 and Figure 5).

### 3.1. RULA Score

No significance was found for the RULA score. A *p* = 0.58 implies that at this level, no difference can be seen between the two conditions (treatment concepts 1 and 2).

### 3.2. Relative RULA Score Distribution

The relative RULA score occurrence could have seven possible values; therefore, the Bonferroni correction was applied with a factor seven resulting in a threshold for significance below 0.007. Hence, no significant difference could be found at that level, with the lowest *p* = 0.14.

### 3.3. RULA Steps Score

In this resolution level, no significant difference was observed. The smallest *p*-value was 0.03, but with Bonferroni correction, this was not significant.

### 3.4. Relative RULA Steps Score Occurrence

With increasing complexity, the resolution increased (Figure 4). The variable set of the relative RULA step scores had already 56 different variables times bins. For overall comparison, the two sets/scores showed a significance after Bonferroni correction (Table 2). One of them was the relative distribution of step 1 (*p* < 0.001) for the right side at score level 4, as shown in Figure 5 paragraph 4. Treatment concept 1 had a significant higher rating of the score level 4. This could be indicative for a slightly higher overall workload for treatment concept 1. This finding was not significant when only the median RULA step scores were considered (Figure 5—level 3).

### 3.5. Relative Angel Distributions

Finally, the highest complexity and highest resolution can be achieved when the relative angle distribution is considered. With 1400 variable times bin values, this has clearly the highest complexity (Table 2). A SPM-based cluster analysis of the waveforms revealed that 4 angles showed a significant difference between treatment concept 1 versus treatment concept 2 after Bonferroni correction was applied (*p* < 0.001). We identified one difference in the shoulder abduction of the right arm (Figure 5—level 5), which was found within one limit range of the RULA scoring system. Therefore, this difference could not be observed in any of the four other RULA-based levels of complexity.

## 4. Discussion

This is the first study that combines different methodological approaches aiming to compute an automated drop-down analysis of an ergonomic risk, which can be utilized in occupational medicine to investigate a specific work process. In this paper, we have shown the seamless transition from a general, high-level summary such as the **RULA score** to a very detailed and sensitive **relative angle distribution**. The five proposed levels enable the investigator to perform a profound investigation on different complexity levels. All five levels are linked to each other, allowing an easy understanding of the main findings and the possibility to analyze small specific differences on a very detailed level. The RULA score gives a summary, an overview, of the overall ergonomic load of a condition. However, this overview is not sensitive enough to differentiate between similar conditions. Hence, four additional levels were implemented, so the researcher has the ability to investigate the details of a condition, or to analyze differences between two conditions. For instance, it is easy to identify general differences in body regions with the third level, the **RULA steps score**.

With increasing complexity and increasing resolution, smaller details were detected in our study. For the **relative RULA step score occurrence** and the **relative angle distribution**, some significant differences between the two conditions were found. For instance, the relative occurrence of the score 3 was significantly different for the RULA step 1, right shoulder. This difference might be related to the significant different cluster of the relative angle distribution (Figure 5—level 5 center). Treatment concept 2 (bright green) had a smaller angle distribution with an ROM around 10° to 30° for the abduction, while this angle ranged from ~5° to 50° for treatment concept 2 (dark green). The distribution showed a significantly different cluster in the range of 40° and 50° abduction, even though it was very variable for different subjects.

The RULA method [8] is a well-established method that allows the investigation of work-related upper limb disorders. This paper is based on the RULA method. The novelty of this paper is the establishment of a code that calculates the RULA score based on 3D inertial motion sensors. This code uses the output of the Xsens motion capture system and calculates the RULA score based on angle ranges of the original RULA publication. The continuous signals allow a deeper investigation of the work-related postures. In order to deal with the increased amount of data, this paper introduces a five-step workflow to picture the data from a condensed high-level view all the way down to the low-level distribution of the individual angles. The five-step workflow consists of the **RULA score** that is a match to the original RULA definition; the **relative RULA score occurrence**; the **RULA step score**, where any of the 15 RULA steps can be selected; the **relative RULA step score occurrences**; and the **relative angle distribution**. This approach was successfully implemented in dentistry, where two specific concepts were investigated.

The **RULA score** gives a general overview of the assessed data. It is a good summary of the overall ergonomic workplace conditions. This score will give a good indication if a workplace is overall ergonomically well designed or if improvements are needed to improve the ergonomic quality. However, the high generalization of this value does not allow a proper investigation of the actual risk factors. The **relative angle distribution** is the counterpart to the **RULA score** with respect to resolution and complexity. Being easily accessible through the IMU units, this measure gives the most detailed information about the actual workplace situation. Even small details can be visualized through the distribution. With this high resolution, small differences between workplace conditions will be reflected right away. This is useful in situations where the differences between conditions are small and do not appear already in the **RULA step scores**. The flexion—extension angle of the right shoulder (Figure 5—level 5) gives an example of such a circumstance. A small significant difference was seen at an angle range of 60°–65°. While in condition 2 (bright green) no angles were measured at this spectrum, there was some occurrence of this angle in condition 1. However, as this difference was entirely located in the score 3 (Figure 5—level 5), this difference might not be reflected in the higher levels of **RULA score** or **RULA step scores**. The main drawback of the **relative angle distribution** is the number of variables. It is hard to get a clear interpretation, since the number of variables is relatively large. One might also get lost in minor details not being able to grasp the large picture.

Automized ergonomic risk assessments based on inertial motion capture and RULA have hardly been published so far. This combined approach was published by Vignais et al. [7] and implemented using the example of physical work steps. This method was further developed in studies with supermarket cashiers [11] and general working tasks [32], which produced results comprising **RULA score**, **relative RULA score occurrence**, an extract of **RULA step scores**, and **relative RULA step score occurrences**. However, for the evaluation of workplace ergonomics, a drop-down approach might be best suited.

The **RULA score** will provide a first very general indication of the potential ergonomic risks of a workplace. However, a differentiation between very similar workplace situations cannot be expected at a small resolution level. With the ability to dissolve seamless step one level down with respect to generalization, it is possible to gain resolution and, therefore, to find possible differences between conditions. These differences might be important to understand the influence of changes through related conditions on the workplace ergonomics and enable the researcher and/or the practitioner to make better decisions, which are based on quantified assessments.

It has to be noted that there are some limitations to the generalization of the code. The code presented in this paper has been designed for assessments of common work processes in dentistry. Therefore, some assumptions have been made that have to be critically evaluated when using this code for studies in other workplaces. These assumptions are presented in Table 1. For instance, it is assumed that dentists do not carry a load of more than 20 Newton with their hands. During common work processes of dentists, this assumption is clearly fulfilled; however, a generalization might not be possible. Furthermore, the focus in our study was set on the evaluation of a sitting position. Hence, this code does not include a proper assessment of situations where the study subject lifts any weights or where loads are picked up from the ground. However, based on the presented MATLAB code, the automated RULA can be modified for any other professional group. Overall, it has been shown that the ergonomic risk assessment profits from detailed high-frequency inertial captured data in the real working environment.

Within the field of dentistry, it should be evaluated if anthropometric differences do make a difference between RULA score assessments. For instance, a smaller arm length might change the shoulder and neck angles, changing the specific RULA scores. The two genders differ in their anthropometric appearance and might therefore result in different RULA scores as well. In addition, the proposed method should be well suited to provide an improved evaluation of the work place configurations. Thus, it might be possible to better understand the potential ergonomic risks of workplaces. For example, applying this method to a larger cohort would provide the opportunity to determine differences in chair configurations. This could ultimately allow for adjustments reducing the ergonomic risks. Furthermore, the wearable IMUs allow an “in the field” assessment of ergonomics. Therefore, future studies should investigate the ergonomic risk in a variety of different work scenarios. These can range from factory workers to skilled worker who face considerable ergonomic demands during their day-to-day work.

## 5. Conclusions

This study reports a novel methodological approach allowing the objective and detailed ergonomic analysis of specific work conditions. It is a benefit of the presented method to switch easily between different levels of complexity in the respective analysis. While first insights can be gained by applying the highest level of generalization onto the dataset, the underlying values measured with the IMU system give further insights. While the RULA score was not sensitive enough in the assessment of the two work conditions, the relative RULA step score and the relative angle distribution were able to show two and four differences, respectively. This additional information will give helpful information when adapting the workplace ergonomics in dentistry or for the interpretation of small but relevant differences in workplace configurations.

## Figures and Tables

**Figure 1 sensors-21-04077-f001:**
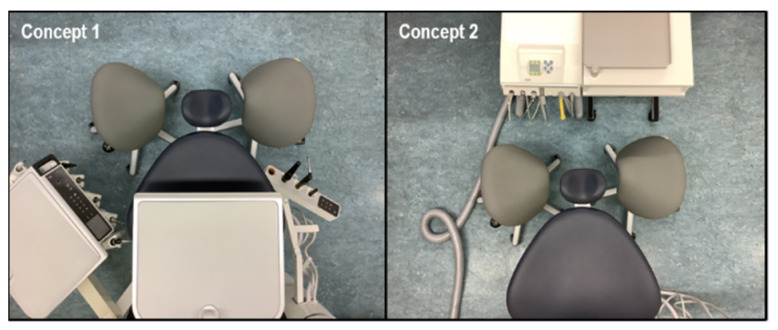
Dental treatment concepts 1 and 2. In both illustrated dental treatment concepts, dentists treat on the left chair (9 o’clock), while dental assistants treat on the right chair (3 o’clock). In treatment concept 1, the functional area is divided, while in treatment concept 2, dentist and assistant share the same functional area.

**Figure 2 sensors-21-04077-f002:**
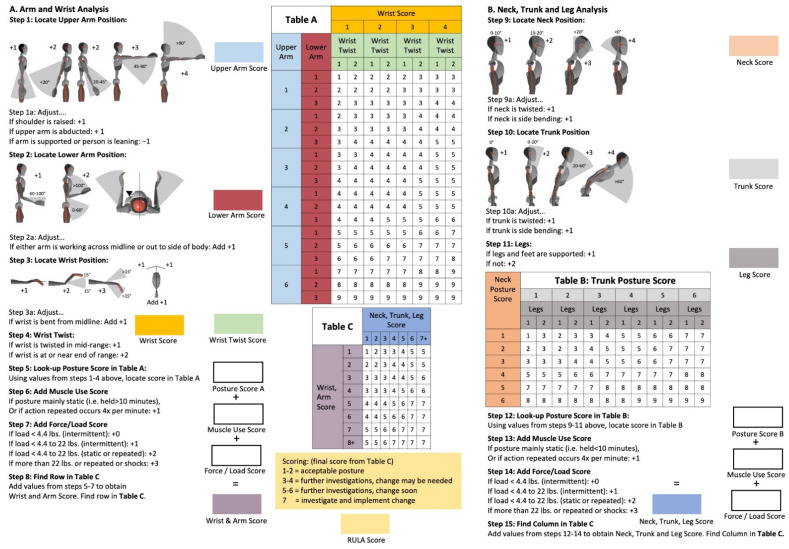
RULA worksheet including all relevant steps.

**Figure 3 sensors-21-04077-f003:**
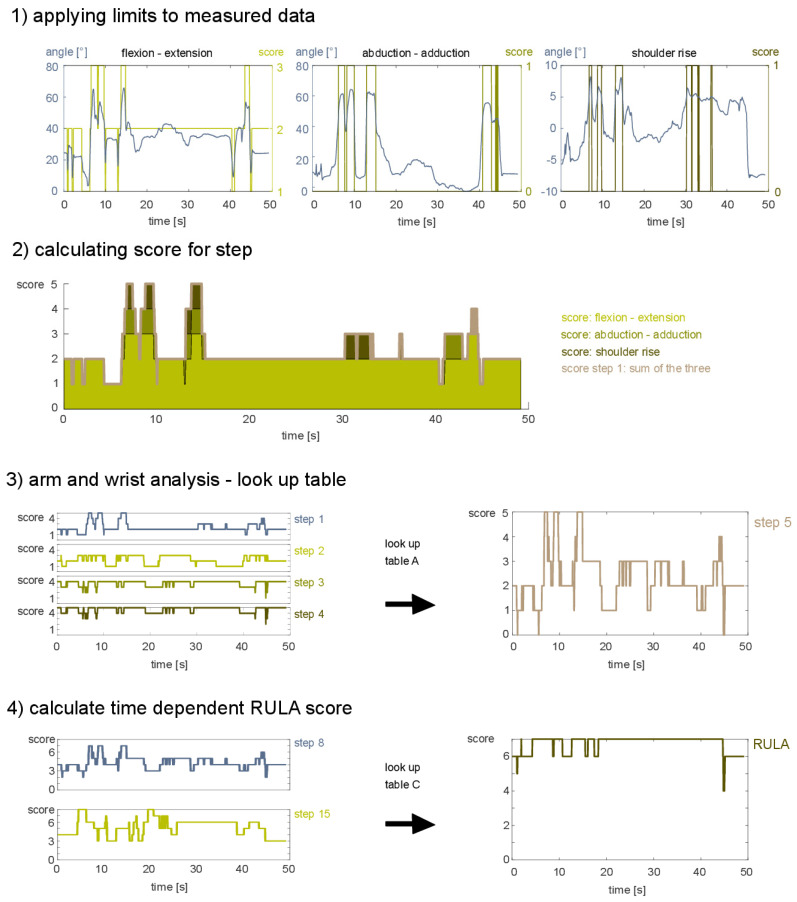
Five main processing stages to calculate the RULA score for time-dependent signals. The displayed processing stages highlight the key features of the workflow (only a small subset of variables is displayed). Raw data 1: the limits for the specific joints are applied for all time points, leading to a downscaling of the resolution. For some steps, more than one joint is used, and the final score for this step is calculated by adding the individual scores at every time point 2. The score for the arm is calculated by a look up table for all time points 3. A similar procedure is used to combine the final score for the arm (step 8) and the final score for the trunk (step 15) for every time point 4. The RULA score is calculated based on the median value of the whole time series 5.

**Figure 4 sensors-21-04077-f004:**
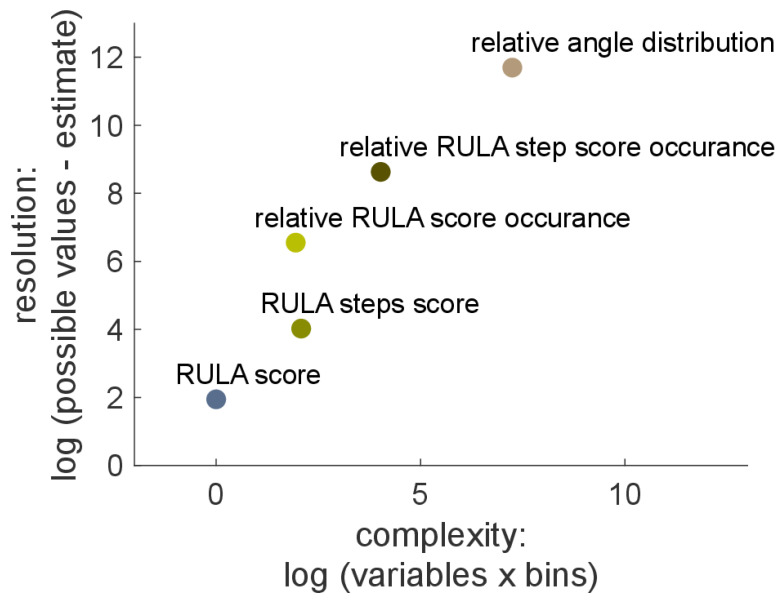
Complexity versus resolution. The complexity was based on the number of variables used in the individual steps. The resolution is based on the number of scores or on the number angles, where 1° bins were used.

**Figure 5 sensors-21-04077-f005:**
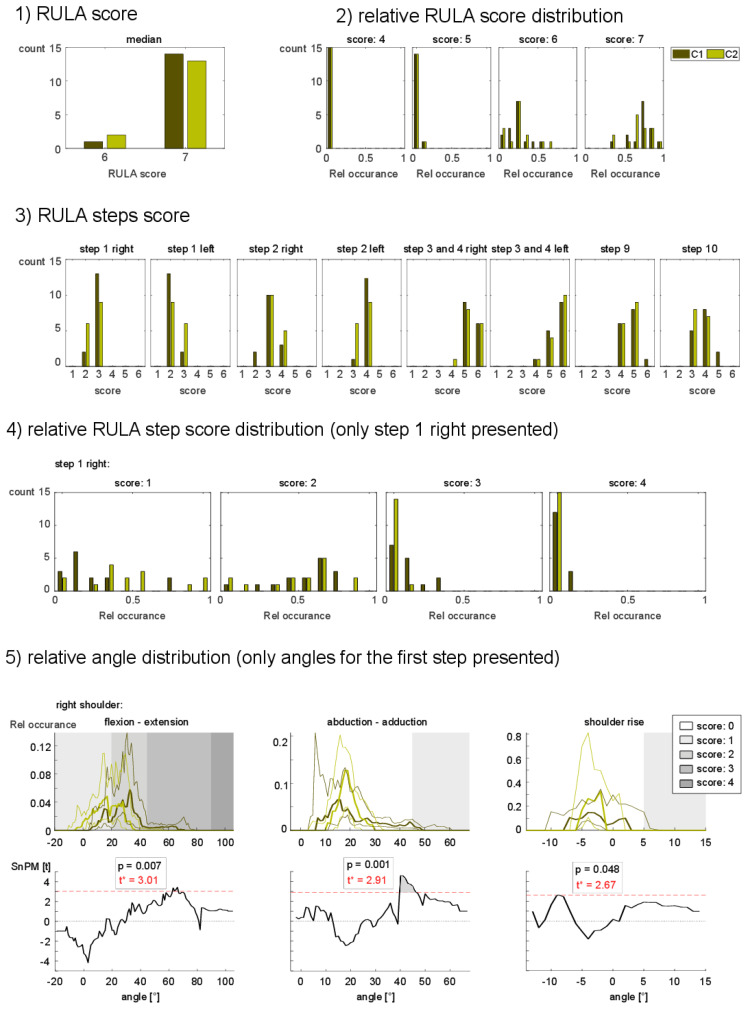
Levels of complexity. The plots show the different metrics calculated from the IMU-based RULA assessment. From top to bottom, the graphs increase the complexity. The RULA score has only one value per condition. C1 indicates treatment concept 1, and C2 indicates treatment concept 2.

**Table 1 sensors-21-04077-t001:** RULA modifications. This table sums up the modifications to several original RULA steps that were applied in order to quantify and automate qualified thresholds.

	Parameters	Modifications of the RULA Parameters
STEP 1	Shoulder raising	The IMU system computed the elevation of the shoulder girdle, thus angles over 5° add +1 to the “shoulder raising score”.
	Upper arm abduction	Abduction angles superior to 45° lead to +1 to the “upper arm score” [7].
	Arm supported	During dental tasks, the arms were not supported so the “arm supported score” was set so 0 [7].
STEP 2	Arm working across midline or out to side of body	We added +1 to the “lower arm score” when the forearm worked across midline or outside of body. These events were computed using the IMU data.
STEP 3	Wrist bending from midline	When the radio-ulnar deviation angle was inferior to −10° (radial deviation) or superior to 10° (ulnar deviation), +1 was added to the “wrist score” [7]. The RULA score does not specify the amount of wrist bend.
STEP 4	Wrist twist	For wrist rotation angles between 45° and −45°, we added +1 to the “wrist twist score”, while rotation angles from 45° to 90° or −45° to −90° lead to +2 to the “wrist twist score”. The RULA score does not specify the amount of wrist bend.
STEP 6	Muscle use score of arm and wrist	Static and dynamic muscle use was estimated based on the time dependency of the joint movement.
STEP 9 + 10	Neck and trunk twist	When the head/trunk rotation angle was inferior to −10° or superior to 10°, +1 was added to the “neck position score”/“trunk position score” [7]. Specific values were added to determine the extent of the movement.
	Locate trunk position	The nomenclature of the trunk position in the sagittal plane was adjusted. Briefly, −5° to 5° lead to +1, +5° to +20° lead to +2, and +20° to +60° or <−5° lead to +3 and >+60° lead to +4.
	Neck and trunk side bending	When the head/trunk angle in the frontal plane was inferior to −10° or superior to 10°, +1 was added to the “neck position score”/“trunk position score” [7]. Specific values were added to determine the extent of the movement.
STEP 11	Legs and feet supported	The “leg score” was fixed to +1 as the dental professionals remained seated during their tasks, and therefore, legs and feet were supported.
STEP 13	Muscle use score of neck, trunk, and legs	Static and dynamic muscle use was estimated based on the time dependency of the joint movement.
STEP 14	Force/load score	This score was fixed to 0, since no weight over 2 kg is lifted in the dental practice.

**Table 2 sensors-21-04077-t002:** Summary of the five levels of complexity. The complexity versus the resolution is also plotted in Figure 4. The complexity is calculated based on the number of variables or the number of variables times bins (whatever is applicable). The resolution is estimated based on the number of possible values. Test statistic was applied, and number of significant findings were carried out. min *p*: the maximal significance that was found.

Parameter set	Complexity Variables Times Bins	Resolution Possible Values-Estimate	Test	Significant Difference	min *p*	Correction	Effective *p* after Bonferroni
RULA score	1	7	Wilcoxon rank sum	No	0.5765	No	
Relative RULA score occurrence	7	700	Wilcoxon rank sum	No	0.1402	Bonferroni	0.007
RULA step score	8	56	Wilcoxon rank sum	No	0.0363	Bonferroni	0.0063
Relative RULA step score occurrence	56	5600	Wilcoxon rank sum	2	<0.001	Bonferroni	0.0014
Relative angle distribution	1400	120,000	SPM	4	<0.001	Bonferroni	0.0021

## Data Availability

The custom written MATLAB code can be found here: https://github.com/ChristianMaurerGrubinger/RULA (accessed on 11 June 2021).

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
