# Peer review of "Combining Ergonomic Risk Assessment (RULA) with Inertial Motion Capture Technology in Dentistry—Using the Benefits from Two Worlds"

_sensors, 2021, doi:10.3390/s21124077_

Round 1

Reviewer 1 Report

In the manuscript, the authors provided an automated rapid upper limb assessment (RULA) with inertial sensors to identify the imbalances between work conditions and the physiological capability of employees. They then proposed five scenarios to check the performance. However, although the authors claim they provided an automated method, the reviewer did not find how the procedure processed automatically. Additionally, the reviewer suggests authors rearrange the structure of context since it is hard to follow the authors' concepts. Some questions are listed as follows.

  1. Please check the grammar.
  2. Please rearrange the structure of the abstract and provide the quantitative results regarding this research work. Meanwhile, please provide a performance comparison between this work and conventional techniques. The authors should exhibit their merits in the relevant fields.
  3. The explanation of the RULA in the introduction section isn’t so straightforward to understand its operational conception. To add a description about RULA will benefit the participants in the relevant field.
  4. Please define the phrase 12m/3f mentioned in lines 18 and 102.
  5. The positions of the dentists and the assistants in fig. 1 are difficult to imagine by the wording. Please rearrange these two scenes. Please delineates the relative positions between the human body and the sensors/cameras. And please define the meaning of the sentence in line 130: “…region 16 o 11 in the first quadrant… .”
  6. Please validate that the simplified five stages, as shown in Fig. 3, can have a similar performance as the original RULA method. Please provide their performance comparison quantitatively.
  7. Please explain how to record and measure the angle ranges of movements as mentioned from lines 192 to 195. By utilizing the inertial sensors or the camera?
  8. Figures 3 and 5 are exactly the same. Please check them.

Author Response

 Reviewer 1

In the manuscript, the authors provided an automated rapid upper limb assessment (RULA) with inertial sensors to identify the imbalances between work conditions and the physiological capability of employees. They then proposed five scenarios to check the performance. However, although the authors claim they provided an automated method, the reviewer did not find how the procedure processed automatically. Additionally, the reviewer suggests authors rearrange the structure of context since it is hard to follow the authors' concepts. Some questions are listed as follows.

  1. Please check the grammar.

A native english speaking colleague revised the language.

  1. Please rearrange the structure of the abstract and provide the quantitative results regarding this research work. Meanwhile, please provide a performance comparison between this work and conventional techniques. The authors should exhibit their merits in the relevant fields.

We have clarified, that the presented method is a clear addition to conventional, manual methods. The conventional RULA methods is based on a manual approach, where in the optimal way a 2D picture of the main working position is used. With this, only a specific position is evaluated. In addition, the drawback of a 2D measurement of angles is well documented (projection errors, perspective errors etc.). The presented method uses all the advantage of the conventional method adding the advantages of a 3D motion capture techniques. Briefly, these advantages are capturing the data allowing the investigator to go back into the data, a 3D measurement free of projection and perspective errors as well as measuring the full task and therefore capturing all movements, any transition from one position into the other etc. We have changed the abstract to clarify this point.

  1. The explanation of the RULA in the introduction section isn’t so straightforward to understand its operational conception. To add a description about RULA will benefit the participants in the relevant field.

Thanks for the hint. We tried to adjust the introduction section about RULA. Our method is based on the RULA method, but the RULA method itself is well documented in the literature. In addition to our brief introduction to the RULA score we have added several references, to allow the reader to easily implement the RULA method themselves. As the focus of the paper was the introduction of several additions to the conventional RULA method, we felt that the provided introduction and the additional references are enough to follow all the specific steps introduced in this article without lengthening the article too much.

  1. Please define the phrase 12m/3f mentioned in lines 18 and 102.

The phrase has been adjusted in both positions. “Fifteen (12 males/3 females)…”

  1. The positions of the dentists and the assistaänts in fig. 1 are difficult to imagine by the wording. Please rearrange these two scenes.

Thank you for the comment. We changed the figure legend of figure 1 accordingly. “In both illustrated dental treatment concepts dentists treat on the left chair (9 o’clock), while dental assistants treat on the right chair (3 o'clock). In treatment concept 1 the functional area is divided, while in treatment concept 2, dentist and assistant share the same functional area.”

Please delineates the relative positions between the human body and the sensors/cameras.

The kinematic analysis was performed using the inertial motion capture system MVN Link from Xsens. All participants wore a lycra suit with sensors attached. These information can be derived from the methods part “Measurement system” (line 165-171).

And please define the meaning of the sentence in line 130: “…region 16 o 11 in the first quadrant… .”

We rewrote the term, in order to make it more comprehensive: “A standardized surgical task was performed in both dental treatment concepts (Fig. 1). The dentists conducted a palatinal and marginal incision in regio 16 to 11 in the first quadrant for approximately 60 seconds.” The terms palatinal, marginal and quadrants are basic dental terms. It basically means, that die subjects cut from tooth no. 16 to 11 in the first quadrant (the mouth is divided in 4 regions).

  1. Please validate that the simplified five stages, as shown in Fig. 3, can have a similar performance as the original RULA method. Please provide their performance comparison quantitatively.

Thank you very much for your question. The presented method is an extension of the classical RULA method in at least two ways. First not only a single posture is measured during a task, but the whole task is recorded. Secondly, in addition to the classical final RULA score, the presented five levels provide additional in depth information.

  1. Please explain how to record and measure the angle ranges of movements as mentioned from lines 192 to 195. By utilizing the inertial sensors or the camera?

Apparently the measurement protocol was not clear enough. The kinematic analysis was conducted using the Xsens motion capture system. The Camera was synchronized in order to be able to allocate measurements, in case mix-ups or other mistakes occur during the measurements. Since it was an iPad, it was solely used to detect mistakes, not for optical motion capture.

We have therefore updated section 2.3 Measurement protocol:

“In order to measure whole body kinematics, all 17 sensors were attached, as specified by the manufacturer. This IMU system was used to measure the 3D angle of the segments and to calculate the 3D joint angles required to calculate the RULA score.”

We changed the sentence accordingly: “In order to have an extern reference, the entire (..)”

And the sentence: “The joint angles are calculated based on the measured angles of the IMUs.” To section 2.7 Implementation of RULA to IMU based evaluation

  1. Figures 3 and 5 are exactly the same. Please check them.

Thank you very much for this utterly devastating point. During the final paper preparation phase we have confused ourselves, resulting in two identical images. We have now added the right figure 5 which is – obviously different from figure 3. Apologies for this mistake.

Reviewer 2 Report

Review

The manuscript presents research related to ergonomic studies of work environments. As a result of the research, the improvement of the traditional RULA risk assessment method was developed and presented by the authors. The significance of this is that the improved method can reveal higher resolution, more accurate kinematic relationships, and risk factors, for example, according to workplace ergonomics. Based on the experimental results reported in the article, the efficiency and applicability of the proposed method are not in question.

In connection with the research, it should be noted that the number of subjects included in the study is low, therefore the sample cannot be considered statistically representative.

My other observation is that the authors indicated in the studies the age distribution of the selected participants, which may suggest that the impact of the work environment is likely to correlate with long-term work, but this was not explained in the manuscript. Related to this is the following observation: in addition to age, significant information may also come from the physique of study participants. It would be worthwhile to examine whether there is a correlation between different body shapes and the standardized work environment presented that significantly influences the outcome of risk assessment (e.g., a low person exhibits quite different characteristics in the outlined work environment than a higher-than-average subject with longer arms). These would, in my view, be aspects that need to be examined in relation to the issue of ergonomics at work.

Overall rating:

The structure of the article meets the requirements of a scientific publication.

The chapters of the manuscript articulate the content well.

The abstract intelligibly summarizes what is described in the article.

Using the literature, the Introduction supports the need for the topic, the theoretical and practical background of the study.

The literature references used in the manuscript are relevant, the amount used is proportional to the work included in the manuscript and the length of the manuscript.

Regarding the topic of the article, I am not sure that Sensor magazine is the right medium for publication. The topic contains much more applied scientific results, the connection with sensors is only that the authors use sensor-based measurement results as source data when developing the improved risk assessment method.

Author Response

The manuscript presents research related to ergonomic studies of work environments. As a result of the research, the improvement of the traditional RULA risk assessment method was developed and presented by the authors. The significance of this is that the improved method can reveal higher resolution, more accurate kinematic relationships, and risk factors, for example, according to workplace ergonomics. Based on the experimental results reported in the article, the efficiency and applicability of the proposed method are not in question.

In connection with the research, it should be noted that the number of subjects included in the study is low, therefore the sample cannot be considered statistically representative.

My other observation is that the authors indicated in the studies the age distribution of the selected participants, which may suggest that the impact of the work environment is likely to correlate with long-term work, but this was not explained in the manuscript. Related to this is the following observation: in addition to age, significant information may also come from the physique of study participants. It would be worthwhile to examine whether there is a correlation between different body shapes and the standardized work environment presented that significantly influences the outcome of risk assessment (e.g., a low person exhibits quite different characteristics in the outlined work environment than a higher-than-average subject with longer arms). These would, in my view, be aspects that need to be examined in relation to the issue of ergonomics at work.

Overall rating:

The structure of the article meets the requirements of a scientific publication.

The chapters of the manuscript articulate the content well.

The abstract intelligibly summarizes what is described in the article.

Using the literature, the Introduction supports the need for the topic, the theoretical and practical background of the study.

The literature references used in the manuscript are relevant, the amount used is proportional to the work included in the manuscript and the length of the manuscript.

Regarding the topic of the article, I am not sure that Sensor magazine is the right medium for publication. The topic contains much more applied scientific results, the connection with sensors is only that the authors use sensor-based measurement results as source data when developing the improved risk assessment method.

Thank you very much for your comments. The purpose of this paper was to present a new method, and provide new concepts to evaluate work space related kinematic data. We have provided an automatic work flow, that allows the measurement and analysis of the upper limb referencing to a well documented, standardized scientific method (RULA). We think, that the 15 subjects demonstrate well the sound implementation of the methods. We do also think that we took care to not over interpret the results. The reviewer mentioned the anatomical variation that exists even if a selection based on a geographical region is used. Indeed, it would be very tempting to analyze the data with respect to anatomical differences. However, the sample size is by far too small, to consider these effects. The presented study is part of a larger study (Ohlendorf D, Maltry L, Hänel J, Betz W, Erbe C, Maurer-Grubinger C, Holzgreve F, Wanke EM, Brüggemann D, Nienhaus A, Groneberg DA. SOPEZ: study for the optimization of ergonomics in the dental practice-musculoskeletal disorders in dentists and dental assistants: a study protocol. Journal of Occupational Medicine and Toxicology 2020;15(1):1-9.), which actually addresses these above mentioned questions with a larger sample size.

Regarding your second point. Since the aim of this study was to create a script that automizes the risk assessment by combining RULA with inertial question and then to compare the resolution of different methods including in that script, the age, gender, body shape or years at work do not influence the results of this study. We compared two very similar conditions in order to show exemplary how important different resolution options in kinematics can be. Thus, we think that in this manuscript further comparisons or aims would exceed the scope of the manuscript.

Reviewer 3 Report

The last sentence of the Introduction "This manuscript focuses on the methodological development of the approach and not 
on the evaluation of the sample data." should be better discussed. This is the original contribution of the paper and it should be better highlighted. Even the Discussion section should better emphasize the original findings of the present paper, when compared to related approaches in the literature. 

It is fairly reported that "there are some limitations to the generalization of the code" and that "the code presented in this paper has been designed for the use in dentistry" How does the paper title represent this scenario? In this respect, the specific advantages of the proposed approach in dentistry should be reported in the whole paper, including the Conclusions.

Author Response

The last sentence of the Introduction "This manuscript focuses on the methodological development of the approach and not on the evaluation of the sample data." should be better discussed. This is the original contribution of the paper and it should be better highlighted. Even the Discussion section should better emphasize the original findings of the present paper, when compared to related approaches in the literature. 

We have added some literature to the discussion (line 360-363): “This approach was further developed in studies with supermarket cashiers [32] and general working tasks [33] including results comprising RULA score, relative RULA score occurrence and an extract of RULA step scores and relative RULA step score occurrences.”

It is fairly reported that "there are some limitations to the generalization of the code" and that "the code presented in this paper has been designed for the use in dentistry" How does the paper title represent this scenario? In this respect, the specific advantages of the proposed approach in dentistry should be reported in the whole paper, including the Conclusions.

The reviewer is right in the aspect that the code has been written to investigate the work place situation in dentistry. The wording of the title and the conclusion has been adapted accordingly. However, it is also important to mention, that the RULA implementation was done in a very general fashion, and therefore the code can be used for any seating position. In addition, only small changes will allow the use of the code to any working task.

Reviewer 4 Report

Please, see the attached PDF file. 

Author Response

General comments

Thank you for the careful reading of the paper. However it seems, that the uploaded version as a slightly different appearance compared with our original version. We did not find any of the mentioned word separation in our text.

We numbered the sections accordingly.

We replaced all spelled out words with their abbreviations.

Thank you for your concern. While we generally like short and concise captions, we felt it useful, that the figures and tables can stand on its own, with only the caption and the drawing or table. However, we managed to reduce the longest caption (caption for figure 3)):

“Figure 3. Five main processing stages to calculate the RULA score from time dependent signals. The displayed processing stages highlight the key features of the workflow (only a small subset of variables displayed). Raw data 1), the limits for the specific joints are applied for all time points, leading to a downscaling of the resolution. For some steps more than one joint is used, the final score for this step is calculated adding the individual scores at every time point 2). The score for the arm is calculated through a look up table 3 for all time points. As similar procedure is used for every time point to combine the final score for the arm (step 8) and the final score for the trunk (step 15) 4). The RULA score is calculated based on the median value of the whole time series 5)”

Keywords

Thank you for the hint. We adjusted the list of keywords accordingly.

Methods

The main purpose for the study was to implement an automated RULA assessment. This was done by the combination of an IMU based measurement system and by an automated post processing work flow. It is our understanding, that the main result, meaning the successful implementation of the method is not affected by the selected of the sample. However, we were very careful to not mentioning any possible results related to anthropometric difference. At the current point we cannot speak about gender specific differences for the two proposed concepts. A larger study, investigating the difference between different working groups, but also between different genders is currently conducted.

We selected to measure joint angles by means of the Xsens motion capture system. This system includes 17 inertial sensors, which are provided by the manufacturer. Since we wanted to capture whole body movements, we used all 17 sensors.

We adjusted the sentence accordingly (line 112-114): “For this purpose, the test persons had to wear a measuring suit. In order to measure whole body kinematics, all 17 sensors were attached, as specified by the manufacturer.”

Implementation of RULA to IMU based evaluation

Thank you for spotting our mistake providing the work figure 5. We have corrected this in the reviewed version. The implementation of the RULA is displayed in figure 2. The purpose of figure 3 is to highlight the fact, that we use time dependent signals, and that we use the full waveform of the signal for our analysis. We have adapted the caption of figure 3 to highlight this fact, and to clarify, that we do not change the general protocol of RULA. Figure 5 illustrates the results for the five proposed levels. It is our understanding, that the comprehensive result shown in figure 5 can be easier understood with the figure 3 in place.

Discussion

We added a comparison to recent similar studies to the discussion (line 360-363). “This approach was further developed in studies with supermarket cashiers [32] and general working tasks [33] including results comprising RULA score, relative RULA score occurrence and an extract of RULA step scores and relative RULA step score occurrences.”

We added the term (383-393): “Within the field of dentistry, it should be evaluated, if anthropometric differences do make a difference between RULA score assessments. For instance a smaller arm length might change the shoulder and neck angles, changing the specific RULA scores. The two gender differ in their anthropometric appearance and might therefore result in different RULA scores. In addition, the proposed method should be used to better evaluate work place configurations. With the method, it might be possible to better understand the potential ergonomic risks of workplaces. Applying this method to a larger cohort it might be possible to determine differences in the chair configurations. This could ultimately allowing adjustments reducing the ergonomic risks.

The wearable IMUs allow an “in the field” assessment of ergonomics. Therefore, future studies should investigate the ergonomic risk  assessment in a variety of different work scenarios. This can range from factory workers to craftsman during their day to day business...”

Conclusion

We have strengthened the conclusion by adding the following sentence: “ While the RULA score was not sensitive enough to the two investigated work places, the relative RULA step score and the relative angle distribution were able to show two and four differences, respectively.”

Round 2

Reviewer 1 Report

Please revise Fig. 2 since the reviewer can find similar figures on the internet. For instance, http://www.kaemart.it/lab-prog-cad/bovisa/cascini/progetti/1314/Colombo-Longo-Panza/analisi-ergonomica-rula.html, https://www.slideshare.net/lucianomarcelooliveira/rula-71067494, https://ergo-plus.com/wp-content/uploads/RULA-A-Step-by-Step-Guide1.pdf, and so forth. The authors should cite this guideline as a reference rather than use it as a figure in the article. 

Author Response

Thank you very much for your advice. We have double-checked, that we have provided the original implementation of the RULA score. Yes, we have included that one. We are aware of several figures demonstrating the very same RULA assessment in more or less similar figures. In addition, we felt, that it is very important to visualize the basic concept of RULA, in order to better understand the later implementation of the five levels. The first review round did support us, as some comments were issued regarding the workflow and implementation. Therefore, we think that a key element is to have this picture directly included in the paper. In order to avoid copyright issues, we have redrawn the original implementation ourselves. As the RULA concept does not change, the image is indeed somehow similar. Therefore, we have added additional references to web pages that show the illustration. Since some of the references link to companies, we cannot guarantee, that the currently available content will be accessible for the public in the future. This is another reason for presenting our own version of the original RULA implementation.

Reviewer 2 Report

Dear Authors!
Thank you for your feedback. I re-examined the supplemented manuscript, took the amendments into account, and thank you for the discussion and conclusion section being supplemented. Thank you for your work. The purpose of the manuscript is understandable, the method and results are suitable for publication.

Author Response

Thank you for your kind feedback. We are happy that all your previous concerns were properly addressed.

Reviewer 3 Report

The authors are kindly required to expand the comparative discussion, while clearly itemizing the novelties of the proposed approach.

Author Response

The spelling was checked and a few small errors were found and corrected.

                We have extended the discussion providing a summary of the novelties of the proposed approach. “The RULA method [9] is a well-established method that allows the investigation of work-related upper limb disorders. This paper is based on the RULA method. The novelty of this paper is the establishment of a code that calculates the RULA score based on 3D inertial motion sensors. This code uses the output of the Xsens motion capture system and calculates the RULA score based on angle ranges of the original RULA publication. The continuous signals allow a deeper investigation of the work related postures. In order to deal with the increased amount of data, this paper introduces a five-step workflow to picture the data from a condensed high-level view all the way down to the low-level distribution of the individual angles. The five-step workflow consists of; the RULA score that is a match to the original RULA definition, the relative RULA score occurrence, the RULA step score, where any of the 15 RULA steps can be selected, the relative RULA step score occurrences and the relative angle distribution. This approach was successfully implemented in dentistry, where two specific concepts were investigated. ”

Reviewer 4 Report

The paper has been improved, but avoid the short title.

Author Response

We have removed the short title.